# Evidence of structural discontinuities in the inner core of red-giant stars

**Mathieu Vrard** [1,2] ✉, **Margarida S. Cunha** [1], **Diego Bossini**[1], **Pedro P. Avelino** [1,3], **Enrico Corsaro** [4] & **Benoît Mosser** [5]

Red giants are stars in the late stages of stellar evolution. Because they have exhausted the supply of hydrogen in their core, they burn the hydrogen in the surrounding shell . Once the helium in the core starts fusing, the star enters the clump phase, which is identified as a striking feature in the color-magnitude diagram. Since clump stars share similar observational properties, they are heavily used in astrophysical studies, as probes of distance, extinction through the galaxy, galaxy density, and stellar chemical evolution. In this work, we perform the detailed observational characterization of the deepest layers of clump stars using asteroseismic data from Kepler. We find evidence for large core structural discontinuities in about 6.7% of the stars in our sample, implying that the region of mixing beyond the convective core boundary has a radiative thermal stratification. These stars are otherwise similar to the remaining stars in our sample, which may indicate that the building of the discontinuities is an intermittent phenomenon.

Asteroseismology, the study of stellar oscillations, provides crucial information on the precise structure of stellar interiors[1]. This research field has known an important development since the launch of the CoRoT[2] and *Kepler*[3] satellites. The study of red-giant stars[4,5] has greatly benefited from asteroseismic data making it possible to disentangle between hydrogen and helium core-burning stars[6]. Since clump stars[7] share similar observational properties[8], they are heavily used in astrophysical studies, as probes of distance[9], extinction through the galaxy[10], galaxy density[11], and stellar chemical evolution[12]. Oscillations in red giants can be characterized through the analysis of the Fourier spectrum of their light curves. Most of the oscillation modes are mixed modes originating from the coupling between acoustic waves, propagating in the stellar envelope, and gravity waves, propagating in the core of the star[13,14]. They therefore probe the entire stellar interior, allowing us to obtain critical information on the core properties of red-giant stars.

The very high photometric precision of the four-year *Kepler* light-curves has enabled the detailed characterization of the oscillation spectra for many stars. These spectra contain information on specific details of their internal structure. Indeed, regions of sharp structural variations inside a star can have a significant impact on the mode frequencies[15]. This occurs when the structural changes take place on scales comparable to, or smaller than the wavelength of the mode, in which case the structural variations are often called glitches. The signature that glitches in the core of red-giant stars imprint on the oscillation mode frequencies was recently theoretically derived[16,17]. According to these developments, core glitches induce a cyclic modulation in the mixed-mode frequencies. The scale and amplitude of the cyclic modulation depend on the location and amplitude of the structural discontinuity, respectively. This signature has been discovered in the oscillation spectrum of KIC 9332840, a red-giant star observed by the *Kepler* satellite[18].

Here we perform a systematic search for core glitches in red-giant stars. A sample of low-mass clump stars with high signal-to-noise *Kepler* light-curves were selected and analyzed to decipher the presence of deviations in their mixed-mode oscillation pattern. We discovered 23 objects showing clear discontinuity signatures we attributed to the mixing processes occurring in the convective core

[1]Instituto de Astrofísica e Ciências do Espaço, Universidade do Porto, CAUP, Rua das Estrelas, 4150-762 Porto, Portugal. [2]Department of Astronomy, The Ohio State University, Columbus, OH 43210, USA. [3]Departamento de Física e Astronomia, Faculdade de Ciências, Universidade do Porto, Rua do Campo Alegre 687, 4169-007 Porto, Portugal. [4]INAF - Osservatorio Astrofisico di Catania, Via S. Sofia 78, I-95123 Catania, Italy. [5]LESIA, Observatoire de Paris, PSL Research University, CNRS, Sorbonne Université, Université Paris Diderot, 92195 Meudon, France. ✉e-mail: vrard.1@osu.edu

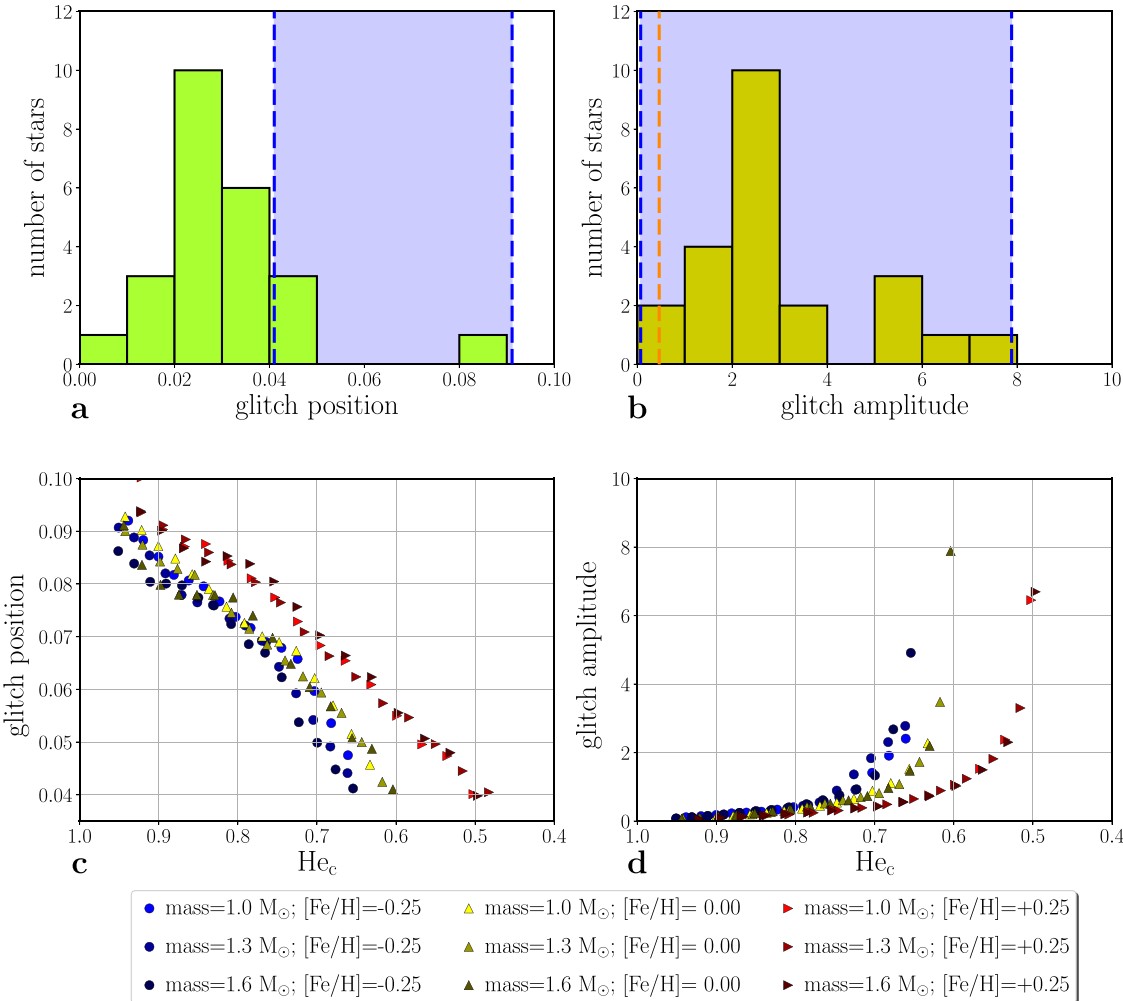

**Fig. 1 | Glitch positions and amplitudes.** Histograms showing the distributions of the dimensionless glitch position ($x^* = \tilde{\omega}_g^*/\omega_g$; **a**) and dimensionless amplitude ($A$; **b**) for the subsample of stars exhibiting glitch signatures. The blue shaded regions indicate the ranges for the corresponding parameters in our sequences of models (see Stellar models description subsection). The latter are also shown, as a function of the core-helium mass fraction ($He_c$) in **c**, **d**. The vertical orange line in **b** indicates the lowest glitch amplitude that was detected for this sample of stars. Source data are provided as a Source Data file.

of clump stars. The detailed research is described in the following sections.

## Results

### Detection and characterization of core glitches

To achieve our goals, we select a sample of 359 red-giant stars with a mass below 1.9 solar masses, previously identified as belonging to the clump[19]. This selection enables us to obtain a sample of stars that follow similar evolutionary tracks during the clump phase, having ignited helium in their core under electron degeneracy, therefore having the same helium-core mass[20]. After the measurement of the global asteroseismic properties and individual mode frequencies of each star in the sample (see Methods subsection Target selection and data preparation for the description of the parameters measurement), we assess the presence of core glitches.

Significant deviations from the usual mixed-mode pattern originating from core glitches are detected in 24 of them (i.e., 6.7% of the sample). We then infer the position ($x^*$) and amplitude ($A$) of the glitches, following the theoretical background developed for interpreting mixed-mode glitches (see Methods subsection Core glitches identification and characterization). These are displayed in panels a and b of Fig. 1. A large dispersion in the two parameters can be noticed (the standard deviation of the position and amplitude distributions correspond, respectively, to 0.014 and 1.79 for a mean

value of, respectively, 0.031 and 3.08), likely reflecting the broad range of metallicity covered by the stars in our sample, as discussed later in this section.

The characteristics of the frequency modulations detected in our sample point towards strong discontinuities located close to the edges of the gravity-waves resonant cavity. According to recent studies[16], there are several regions of significant structural variation in the core of red-giant stars, susceptible of producing this type of modulation. One of them is the hydrogen-burning shell, at the outer boundary of the helium core. Nevertheless, for the mass range considered here, during the clump phase this region is too broad to be seen by the waves as a glitch[16]. Another region is that associated with the chemical discontinuity produced by the first dredge-up[21] that occurs at the bottom of the red-giant branch, when the convective envelope of the low-luminosity red-giant starts to deepen into the star before retracting back. However, for low-mass stars, this discontinuity is smoothed out before the star reaches the clump phase during the so-called luminosity bump[22]. The helium flash[4,23], occurring at the time of the first ignition of helium, could be another candidate to produce a chemical discontinuity in the stellar core. However, clump models do not seem to show a modulation in their oscillation spectrum that can be related to this discontinuity[24]. The produced chemical discontinuities have indeed very small amplitudes as shown by Fig. 2. One last significant structural variation is that associated with the transition

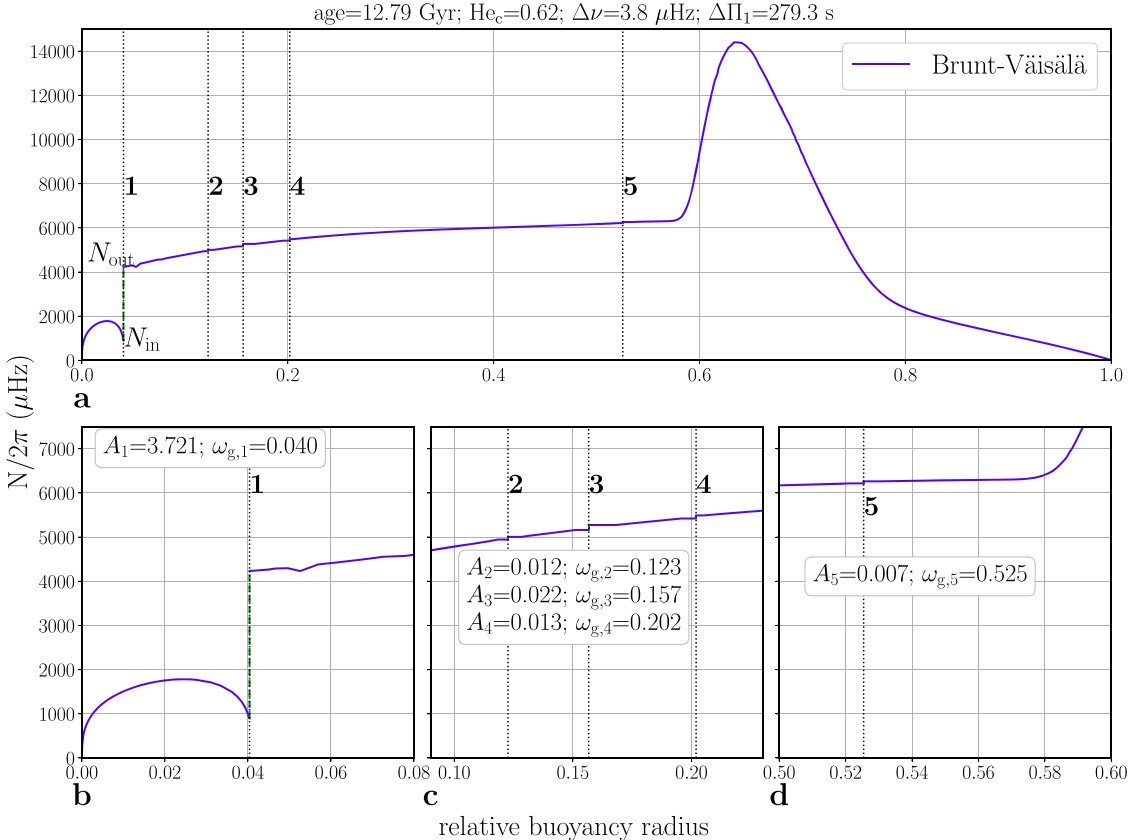

**Fig. 2 | Brunt-Väisälä frequency as a function of the relative buoyancy radius** ($x = \widetilde{\omega}_g^r/\omega_g$) **for one of the computed models (in violet).** The origin of the abscissa marks the inner edge of the g-mode cavity. **a** shows the complete radiative cavity and **b**, **c**, **d** highlight specific parts. The black dotted line labeled 1 (in **a**, **b**) corresponds to the glitch position ($x^* = \widetilde{\omega}_g^*/\omega_g$) due to the chemical discontinuity caused by matter penetration in the radiative cavity and the dashed green line displays the jump in the Brunt-Väisälä frequency at the discontinuity position. The black dotted lines labeled 2 to 5 (in **a**, **c**, **d**) correspond to the glitch position caused by the different helium subflashes. Source data are provided as a Source Data file.

between the convective and the radiative part of the core[16]. Indeed, as a result of the energy released by the fusion of helium, the innermost layers of clump stars are convectively unstable. At the edge of the convectively unstable region, matter penetrates inside the radiative layers, thus inducing chemical mixing but also a chemical discontinuity[25,26]. This discontinuity is located near the inner edge of the gravity-waves resonant cavity and is likely to be responsible for the observed signatures.

## Comparison with models

To explore this scenario further, we considered nine evolutionary sequences of stellar models, covering a range of masses and metallicities, from the onset of helium burning up until the onset of semiconvection, approximately half way through the clump evolution phase. The mixing beyond the convectively unstable core was treated assuming a step-function overshoot with a radiative temperature gradient in the overshoot region (see Methods subsection Stellar models description for a complete description of the models). The full chemical mixing in the innermost layers generates a chemical discontinuity at the edge of the overshoot region which leads to a discontinuity in the Brunt-Väisälä frequency. When the temperature gradient beyond the convectively unstable core has radiative properties, as assumed in our models, the discontinuity affects the propagation cavity of the mode[25] This is in contrast with models where the extra mixing is treated assuming an adiabatic temperature profile. In that case, the discontinuity is placed at the inner edge of the propagation cavity of the mode and does not leave a glitch signature on the mode frequencies.

The range of positions and amplitudes of the glitches present in our sequences of clump-star models are shown by shaded regions in panels a and b of Fig. 1 and as a function of core-helium mass fraction in panels c and d. The glitches of very small amplitude (below the observational threshold) are found in models at the beginning of the clump phase, when the chemical discontinuity remains small. For the remaining models, the glitch positions and amplitudes are within the observed ranges, and vary according to the model metallicity, confirming that the observed signatures can be associated to a discontinuity close to the border of the convectively unstable core, with the spread in the glitch properties resulting from a spread in the stellar metallicities and ages. The fact that the model-glitch positions are found to be biased towards high values of $x^*$ can be understood by considering that the glitch position decreases with evolution (cf Fig. 1c) and our models are evolved only up until the onset of semiconvection.

## Inferring physical information

The panel a of Fig. 3 compares the distribution of period spacings for the stars where glitch signatures are detected with those for the whole sample. The absence of glitches in stars near the lower limit of the period spacings between gravity modes, $\Delta\Pi_1$, is expected because most of those stars just started the evolution on the clump phase and, thus, not enough time has elapsed in their case to generate glitches with amplitudes above the observational threshold. The minimum period spacing of the gravity modes for which signatures of glitches are observed is 246s (to be compared to 229s at the start of the clump phase in our sample) thus providing

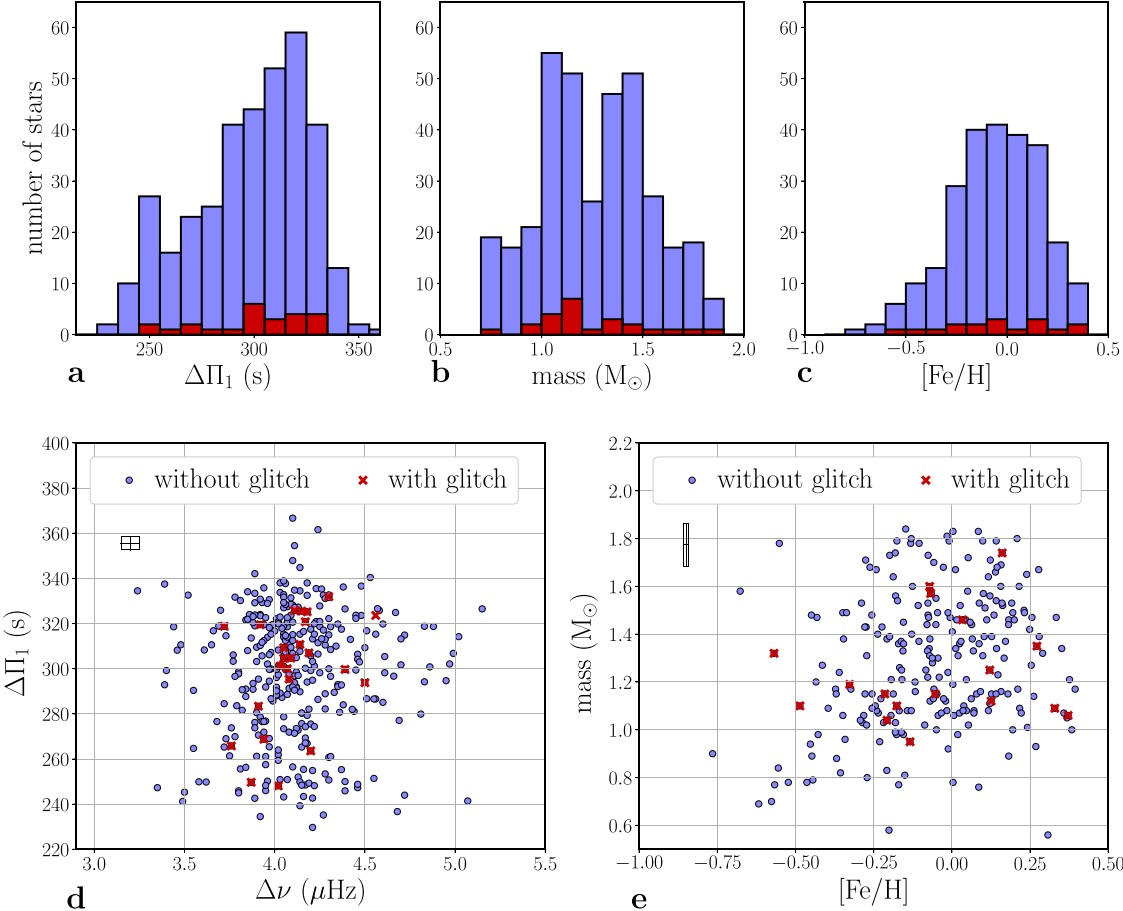

**Fig. 3 | Properties of stars with and without detected glitches.** Histograms representing the number of stars as a function of the period spacing ΔΠ₁ (**a**), mass (**b**) and metallicity (**c**) present in the full sample (in blue) and the stars for which a discontinuity was found (in red). Period spacing as a function of the large frequency separation (**d**) and stellar mass as a function of metallicity (**e**) for the total analyzed sample (blue circles) and the stars for which a discontinuity was found (red crosses). Plots requiring metallicity data show only the subset of stars in the APOKASC catalog. The black squares in the top left corner of **d** and **e** correspond to the parameters median uncertainties which were computed with the fitted techniques referred in method subsection Target selection and data preparation. Source data are provided as a Source Data file.

information about the timescale needed to build chemical discontinuities.

The mixing beyond the border of the convective core in clump stars is a matter of significant debate[25,27]. The detection of core-glitch signatures in a non-negligible number of stars in our sample supports the predictions of models that preserve the radiative properties of the layers adjacent to the convectively unstable core, as suggested for stars at the beginning of the clump phase[24]. Moreover, it rules out models employing chemical mixing prescriptions that do not allow the building of significant structural variations near the core border of the radiative region at any point during the clump phase, such as those assuming an adiabatic thermal stratification. Despite this, the fact that core-glitch signatures are detected in approximately one out of eight clump stars only, raises an important question concerning the physical origin of this dichotomy. The assumptions behind the radiative and adiabatic treatments of the thermal stratification in the extra mixing region are mutually exclusive (one would not expect both to be at play in clump stars). Therefore, differences in the thermal stratification cannot be evoked to explain this dichotomy. Inspection of the signal-to-noise ratio for the stars exhibiting glitches shows that the quality of the data for these stars is similar to that of the whole sample, also ruling out an observational origin for the observed dichotomy. A comparison of the distribution of the period spacings for the subsample of stars exhibiting glitch signatures with that for

the whole sample (Fig. 3a) shows that the two spread similarly the bulk range of period spacings. Hence, an evolutionary origin of this dichotomy can also be ruled out. Likewise, the distributions of mass and metallicity and seismic properties of the two samples (Fig. 3b, c, d, e) are similar, indicating that there is nothing unique about the properties of this subsample of stars. Consequently, one is led to consider two possible scenarios for the observed dichotomy: either the glitches are present throughout evolution in all clump stars, but generally their amplitude is below the observational threshold and what we observe are the largest values of the amplitudes, or core glitches are temporarily smoothed out by some unknown physical process that leads to intermittent changes in the structure of the core along the red-clump evolution. The first scenario is not supported by our models, which predict that glitch amplitudes similar to those observed are common, while the second scenario evokes some physical process not yet included in the models. In either case, the potential impact of these results on state-of-the-art models is unequivocal.

This work opens a window into the characterization of the physical processes taking place inside the core of red-giant clump stars, setting important constraints on state-of-the-art models. Future improvements of these models, based on the constraints discussed here, should significantly improve the characterization of clump stars, strengthening their use in research fields where they play a major role, such as galactic archaeology.

## Methods

### Theoretical description of the influence of a discontinuity on the mixed-mode pattern

Asymptotically, the gravity modes are regularly spaced in period, following the asymptotic period spacing $\Delta\Pi_\ell$[28,29], $\ell$ being the mode degree. Gravity waves propagate only in regularly stratified media, which, in the case of red giant stars, corresponds to the radiative part of their core. Consequently, for these stars, gravity modes are not directly observable. However, a coupling occurs between gravity (g-) and pressure (p-) waves, producing the so-called mixed-modes in red giant star spectra. The precise description of the mixed-mode frequencies as a function of the stellar properties has been previously developed[28,29] assuming a smooth variation of the stellar structure. However, the asymptotic expansion used in these works does not take the presence of structural discontinuities (glitches) into account. Recent works have incorporated the impact of glitches on the dipolar ($\ell = 1$) gravity-mode period spacing ($\Delta\Pi_1$) of red-giant stars, showing that they produce significant distortions in the period spacing pattern[16,17]. In this article, we adapted the formalism in order to obtain the mixed-mode frequency pattern of a clump star with a glitch in its core. The eigenvalue condition in the presence of a glitch and coupling between gravity and pressure modes was demonstrated to be given by ref. [16]:

$$\int_{r_1}^{r_2} k_r \, \mathrm{d}r = \pi\left(n - \frac{1}{2}\right) - \varphi - \Phi, \tag{1}$$

where $k_r$ is the radial wavenumber and $r_1$ and $r_2$ are the limits in the radial coordinate of the g-mode cavity. Moreover, $n$ is a positive integer, and $\Phi$ is the glitch frequency-dependent phase[17] representing the glitch influence on the mode frequencies. The phase $\varphi$ represents the coupling influence on the mode frequencies and can be expressed as a function of the p-wave phase: $\varphi \approx \arctan\left(q/\tan\left(\pi\left(\frac{\nu - \nu_{n,\ell=1}}{\Delta\nu}\right)\right)\right)$, where $\nu$ is the mixed-mode frequency, $\Delta\nu$ corresponds to the mean frequency difference between consecutive pressure modes of same angular degree (also called large frequency separation), $q$ is the coupling parameter between p- and g-waves and $\nu_{n,\ell=1}$ represents the pure pressure $\ell = 1$ mode frequencies.

Next, following previous works[30], we can write $k_r = k_r^0 + \delta_k$ where $k_r^0$ represents the radial wavenumber without accounting for any glitch or coupling and $\delta_k$ corresponds to the perturbation due to the coupling and glitches. The eigenvalue conditions in the Cowling approximation, in the limit of no coupling and without a glitch, translates to[31] $\int_{r_1}^{r_2} k_r^0 \, \mathrm{d}r = \pi\left(n - \frac{1}{2}\right)$. Moreover, considering the definitions of the asymptotic period spacing and of the radial wavenumber inside the g-mode cavity[16], $\delta_k$ can be written as: $\int_{r_1}^{r_2} \delta_k \, \mathrm{d}r = \frac{\pi}{\Delta\Pi_1}\left(\frac{1}{\nu} - \frac{1}{\nu_g}\right)$, where $\frac{1}{\nu_g} = \Delta\Pi_1\left(n_g + \frac{1}{2} + \varepsilon_g\right)$ corresponds to the gravity mode periods, with $n_g$ and $\varepsilon_g$ representing, respectively, the gravity mode radial orders and the gravity phase offset. It is thus possible to write

$$\frac{\pi}{\Delta\Pi_1}\left(\frac{1}{\nu} - \frac{1}{\nu_g}\right) = -\varphi - \Phi. \tag{2}$$

After substituting the coupling phase $\varphi$ in Eq. (2), we find

$$\nu = \nu_{n,\ell=1} + \frac{\Delta\nu}{\pi}\arctan\left(q\tan\left[\pi\left(\frac{1}{\nu\Delta\Pi_1} - \varepsilon_g\right) + \Phi\right]\right), \tag{3}$$

which is close to the expression derived for the mixed-mode frequency pattern without a structural discontinuity[29], having only the addition of the glitch phase. $\Phi$ can be expressed in different ways, depending on the assumed form of the structural discontinuity[17]. Here, for the sake of simplicity, the glitch is modeled by a step-function. This glitch model has the advantage of being described by only 3 free parameters. This choice is also justified by the fact that the frequency range where

oscillation modes are detected is too small to observe the amplitude variation we see for other glitch models[17]. For this specific case, it has been shown that the glitch phase $\Phi$ is given by ref. [17]

$$\Phi = \mathrm{arccot}\left[-\frac{2 + 2A\cos^2(\widetilde{\beta}_2)}{A\cos(\widetilde{\beta}_1)}\right], \tag{4}$$

where $\widetilde{\beta}_1 = 2\frac{\widetilde{\omega}_g^*}{2\pi\nu} + 2\varphi + 2\varepsilon$ and $\widetilde{\beta}_2 = \frac{\widetilde{\omega}_g^*}{2\pi\nu} + \frac{\pi}{4} + \varphi + \varepsilon$. Here, $\widetilde{\omega}_g^*$, $A$ and $\varepsilon$ represent the glitch parameters, namely, the buoyancy radius at the glitch position in the radiative cavity, the glitch amplitude and the phase value, respectively. The buoyancy radius measures the distance from the inner edge of the g-mode propagation cavity, and is defined in terms of the Brunt-Väisälä frequency, $N$, as $\widetilde{\omega}_g^r = \int_{r_1}^{r}[l(l+1)N/r]\mathrm{d}r$. Its value at the glitch position $r = r^*$ is denoted by $\widetilde{\omega}_g^*$, as noted above. Of interest is also the relative buoyancy radius defined here as $x = \widetilde{\omega}_g^r/\omega_g$, where $\omega_g \approx \frac{2\pi^2}{\Delta\Pi_1}$ is the total buoyancy radius[32] of the g-mode cavity, as well as its value at the glitch position, namely, $x^* = \widetilde{\omega}_g^*/\omega_g$.

### Target selection and data preparation

The red-giant star sample was selected among the clump stars observed by the *Kepler* satellite for which the signal-to-noise ratio is sufficient to measure the period spacing $\Delta\Pi_1$[19] and with masses below $1.9\,M_\odot$, in order to guarantee similar helium core masses at the start of the helium burning phase and, therefore, similar evolution during the clump phase. This selection ensures that the variation of $\Delta\Pi_1$ during the clump phase is approximately the same for all sample stars. Among those, since the sample of stars was still too large to analyze object by object, we selected the first 359 stars in the sample by KIC number. Their seismic mass and radius were derived from the measurement of the global seismic parameters, using an autocorrelation technique[33], refined by the use of the pressure mode pattern[34], and the seismic scaling relations[35,36]. The granulation background parameters, corresponding to the signature of stellar granulation, were also estimated with a Bayesian fitting technique[37,38]. For each star, we selected the portions of their spectrum where mixed modes are present. To do so, we identified the radial and quadrupole pressure mode frequencies by the use of the pressure mode pattern[34] and we suppressed the portion of the star spectra around those modes with a frequency width corresponding to three times the width at half-maximum of the modes. Then, we smoothed the power density spectrum and identified the mixed modes as the local peaks with heights above a threshold corresponding to the rejection of the pure noise hypothesis with a confidence level of 99.9%. The presence of rotational splittings was also checked by identifying the number of peaks above the threshold in the smoothed power density spectrum for each mode. If only one peak is present, the presence of rotational splittings is not accounted. After this, the identified modes were fitted using a Bayesian method[37,39] with two different types of peak profiles. Due to the long lifetime of mixed modes, it is possible that the observed modes are not resolved in *Kepler* spectra. In this case, the oscillation peak profile is represented as a sinus cardinal function[40], otherwise, the adopted profile is represented by a Lorentzian[41]. We analyzed the number of frequency bins belonging to the peaks that are above 8 times the background level, thus corresponding to the presence of a signal with a confidence level of 99.9%, in order to determine whether the different peaks we located correspond to resolved modes. The corresponding mode is considered as being resolved, if more than one frequency bin reaches this level for a specific peak, otherwise, it is identified as being unresolved. An example of the mode identification and the fitting results concerning the mixed-mode frequencies is shown in Supplementary Fig. 1 for the star KIC3544063. The stars' metallicities were selected from the DR16 version of the APOKASC catalog[42]. This catalog is not yet complete for *Kepler* objects, therefore we were able to retrieve the metallicities for only 246 objects out of the 359 stars in our sample. A

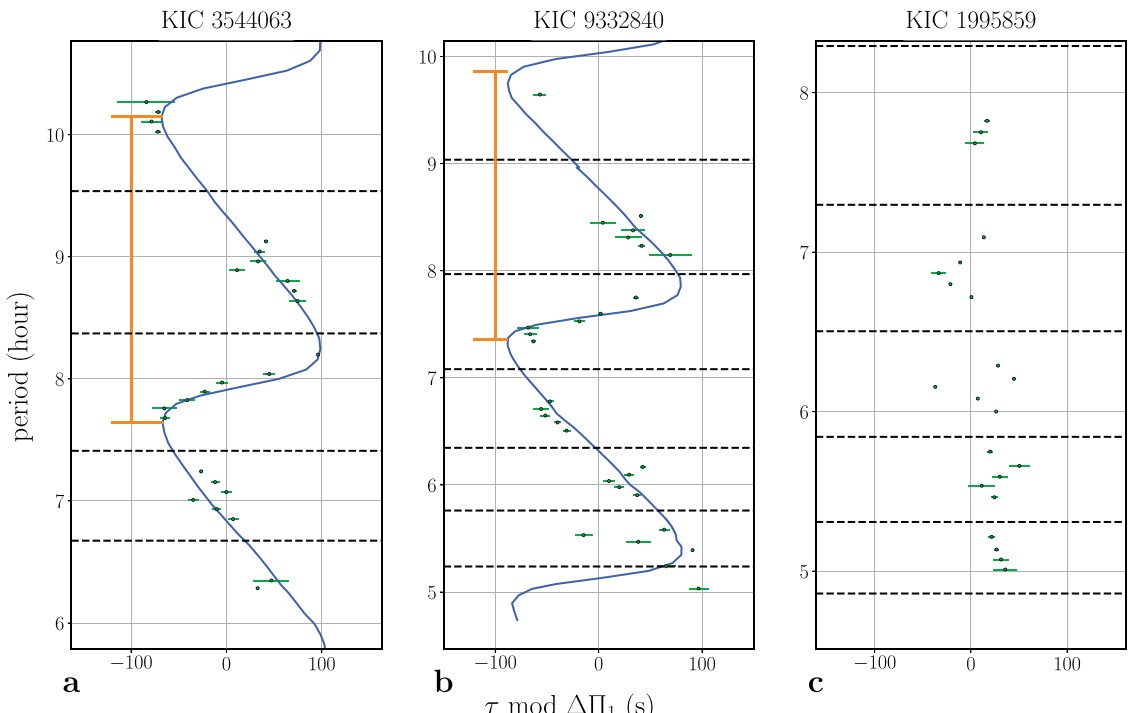

**Fig. 4 | Échelle diagrams for three stars in our sample.** Observed mixed mode periods (green dots) as a function of the stretched periods ($\tau$) modulo $\Delta\Pi_1$ in an échelle diagram for the stars KIC3544063 (**a**), KIC9332840 (**b**) and KIC1995859 (**c**). The uncertainties were computed following the fitting technique described in method subsection Core glitches identification and characterization. The blue line on **a** and **b** represents the best fitted model to the observed mixed mode pattern, the orange lines mark intervals of the inferred signature period ($\tau_{sig}$) given in Table 1 and the dashed-black lines show the frequency positions of the radial ($\ell = 0$) modes. The star KIC1995859 (**c**) shows no evidence of a glitch. Source data are provided as a Source Data file.

summary of the complete sample star's properties is present in Supplementary data 1.

## Core glitches identification and characterization

Mixed modes are observed in the spectra of red giant stars instead of pure gravity modes. In order to remove the pressure component present in the mixed modes, thus isolating the glitch effect on the mixed-mode frequency pattern, we modified the mode frequencies following a change of variable[18,19,30]. The modified (so-called stretched) periods that were obtained through this procedure are called $\tau$. The stretched mixed-mode periods would be regularly spaced in period, if the original mixed-mode pattern had no deviation from the asymptotic mixed-mode pattern[28,29]. An example of that behavior is shown for the star KIC1995859 on panel c of Fig. 4. It is therefore possible to identify the deviations with respect to this regular mixed-mode pattern resulting from the impact of a core glitch, in the modified oscillation spectra. We measured the frequency deviation from the regular mixed-mode pattern for each object in our sample. If the deviations were found significant with respect to the uncertainties on the mode frequencies, the star was considered to be a candidate for presenting a glitch. After this, each star was analyzed manually to see if a change of parameters ($\Delta\Pi_1$,q,$d_{01}$) or the addition of rotation could provide an alternative explanation to the observed frequency deviations. The fitting of the global mixed-mode parameters can indeed converge towards local minimas (aliases[19]), therefore this possibility had to be checked. $d_{01}$ represents here the small separation for $\ell = 1$ modes that is obtained with the universal pattern[34]. The parameter space that was investigated was comprised between $\Delta\Pi_1 = [0.9\Delta\Pi_1 : 1.1\Delta\Pi_1]$, q = [0.05:0.5] and $d_{01} = [d_{01} - 0.05\Delta\nu : d_{01} + 0.05\Delta\nu]$; with a step of 0.1s, 0.05 and $0.01\Delta\nu$ respectively. When no other alternative solution was found, the presence of a glitch was considered to be a valid solution. Among the 359 stars we analyzed, 24 of them showed clear cyclic deviations from the expected regular spacing with an amplitude higher than a tenth of

$\Delta\Pi_1$. For each of these objects, we fitted the observations with the analytical model previously described by Eq. (3) with a glitch phase model corresponding to a step-function discontinuity[17]. A step-function model is suitable when considering these chemical discontinuities (as illustrated in panel b of Fig. 2). The main impact of considering a different glitch model would be seen on the frequency dependence of the glitch amplitude[17]. However, the frequency range where we detect mixed modes in the spectrum of the analyzed stars is too small for those differences to be seen. Based on this model, we first computed the mixed-mode frequency pattern in the presence of a core glitch and then performed the change in variable to obtain the model stretched periods ($\tau$). This ensures that the model mixed-mode frequencies are treated in the same way as the observed mixed-mode frequencies, avoiding any possible bias inherent to this variable change. Then, the model stretched periods, computed for different values of the three model parameters (the glitch amplitude, position inside the resonant cavity and phase) are fitted to the observed mixed modes using the python module *emcee*[43], implementation of the affine-invariant ensemble sample for Markov Chain Monte Carlo (MCMC) fitting technique. The model parameters are then estimated from the best fit solution and the error bars are taken as the 1-$\sigma$ marginalized distribution around this best solution. Examples of the fit performed on the star KIC3544063 and KIC9332840 are shown on panels a and b of Fig. 4. The results for the full sample can be seen on Supplementary Fig. 2. The fitted mixed-mode frequencies for these stars can also be found in Supplementary data 2. We obtained reliable fits for 23 stars among the 24 selected. We discarded the results for the star KIC2156988, despite its mixed-mode pattern being consistent with the presence of a glitch with a very long period and high amplitude. For this star, the observed deviation could indeed also be explained by small variations of the mixed-mode frequencies at high frequencies where the frequency position measurement is less accurate. The results of the glitch parameters fitting for the other 23 stars are given in Table 1.

**Table 1 | Properties of the stars exhibiting a core-glitch signature**

| KIC | Δν μHz | ΔΠ₁ s | q | x* | A | ε | τ_sig h |
|---|---|---|---|---|---|---|---|
| 1724879 | 4.50 ± 0.07 | 293.8 ± 3.5 | 0.22 ± 0.09 | 0.049 ± 0.002 | 0.629 ± 0.473 | 3.112 ± 0.458 | 1.7 ± 0.1 |
| 1726211 | 3.72 ± 0.10 | 322.0 ± 3.9 | 0.36 ± 0.08 | 0.018 ± 0.003 | 2.315 ± 0.293 | −1.460 ± 0.234 | 5.0 ± 1.2 |
| 2303367 | 4.05 ± 0.05 | 307.2 ± 3.2 | 0.31 ± 0.01 | 0.029 ± 0.001 | 2.174 ± 0.550 | 0.036 ± 0.320 | 2.9 ± 0.1 |
| 2442559 | 4.56 ± 0.08 | 323.6 ± 4.4 | 0.33 ± 0.08 | 0.043 ± 0.002 | 0.461 ± 0.114 | −0.412 ± 0.339 | 2.1 ± 0.1 |
| 2448679 | 4.30 ± 0.06 | 330.2 ± 4.2 | 0.32 ± 0.08 | 0.018 ± 0.001 | 2.919 ± 0.261 | 0.477 ± 0.072 | 5.1 ± 0.3 |
| 2573092 | 4.08 ± 0.09 | 295.8 ± 3.1 | 0.29 ± 0.07 | 0.032 ± 0.001 | 2.688 ± 0.148 | 0.011 ± 0.001 | 2.6 ± 0.1 |
| 2583651 | 3.94 ± 0.50 | 280.2 ± 2.4 | 0.29 ± 0.04 | 0.023 ± 0.001 | 7.292 ± 0.610 | 0.016 ± 0.007 | 3.4 ± 0.2 |
| 2714397 | 4.15 ± 0.07 | 325.7 ± 3.5 | 0.39 ± 0.08 | 0.025 ± 0.001 | 2.936 ± 0.721 | −0.877 ± 0.140 | 3.6 ± 0.2 |
| 2995656 | 4.19 ± 0.06 | 306.1 ± 3.1 | 0.29 ± 0.03 | 0.032 ± 0.001 | 3.453 ± 0.663 | −0.267 ± 0.114 | 2.7 ± 0.1 |
| 3112405 | 3.87 ± 0.03 | 248.1 ± 2.0 | 0.21 ± 0.05 | 0.031 ± 0.002 | 2.158 ± 0.492 | 1.135 ± 0.320 | 2.2 ± 0.2 |
| 3117024 | 4.02 ± 0.07 | 248.0 ± 2.2 | 0.15 ± 0.05 | 0.025 ± 0.001 | 2.581 ± 0.522 | 0.953 ± 0.174 | 2.8 ± 0.1 |
| 3120004 | 4.09 ± 0.09 | 303.3 ± 3.0 | 0.25 ± 0.01 | 0.027 ± 0.001 | 1.586 ± 0.5506 | −0.624 ± 0.219 | 3.1 ± 0.1 |
| 3218973 | 4.20 ± 0.05 | 263.6 ± 2.5 | 0.21 ± 0.08 | 0.025 ± 0.001 | 5.803 ± 1.099 | −1.135 ± 0.370 | 2.9 ± 0.1 |
| 3427629 | 3.76 ± 0.10 | 265.9 ± 2.2 | 0.15 ± 0.04 | 0.012 ± 0.001 | 1.951 ± 0.455 | 0.023 ± 0.047 | 6.2 ± 0.6 |
| 3544063 | 4.18 ± 0.05 | 325.2 ± 3.6 | 0.39 ± 0.07 | 0.036 ± 0.001 | 5.097 ± 0.542 | 0.277 ± 0.043 | 2.5 ± 0.1 |
| 3626776 | 4.03 ± 0.06 | 317.3 ± 2.9 | 0.39 ± 0.04 | 0.026 ± 0.001 | 5.152 ± 0.653 | −0.001 ± 0.031 | 3.4 ± 0.2 |
| 3629335 | 4.14 ± 0.06 | 310.2 ± 3.0 | 0.35 ± 0.09 | 0.043 ± 0.001 | 3.757 ± 0.901 | −0.166 ± 0.051 | 2.0 ± 0.1 |
| 3642873 | 4.11 ± 0.03 | 300.7 ± 3.4 | 0.31 ± 0.09 | 0.025 ± 0.001 | 2.492 ± 0.776 | −0.114 ± 0.023 | 3.3 ± 0.2 |
| 3646645 | 3.92 ± 0.07 | 320.5 ± 3.3 | 0.25 ± 0.05 | 0.024 ± 0.001 | 2.858 ± 0.497 | −0.367 ± 0.060 | 3.7 ± 0.2 |
| 3842450 | 4.17 ± 0.05 | 321.8 ± 3.6 | 0.31 ± 0.06 | 0.027 ± 0.003 | 1.532 ± 0.200 | −0.261 ± 0.025 | 3.3 ± 0.4 |
| 3864171 | 4.07 ± 0.25 | 280.2 ± 2.6 | 0.27 ± 0.05 | 0.035 ± 0.005 | 1.643 ± 0.344 | 0.305 ± 0.095 | 2.2 ± 0.3 |
| 3946270 | 4.05 ± 0.05 | 304.9 ± 3.1 | 0.30 ± 0.01 | 0.085 ± 0.001 | 2.676 ± 0.291 | −0.330 ± 0.097 | 1.0 ± 0.1 |
| 9332840 | 4.39 ± 0.08 | 300.0 ± 3.5 | 0.25 ± 0.01 | 0.034 ± 0.005 | 6.604 ± 1.086 | 0.498 ± 0.298 | 2.5 ± 0.4 |

Δν is the large frequency separation, ΔΠ₁ the gravity-mode period spacing, q the coupling parameter. The relative buoyancy radius $x^* = \widetilde{\omega}_g^* / \omega_g$, amplitude A, and phase ε are the dimensionless parameters inferred from the fitting of the glitch-induced modulation, and $\tau_{sig} = \Delta\Pi_1 / x^*$ is the period of the glitch signature. The glitch parameters $x^*$ and A are directly related to the position and amplitude of the discontinuity, respectively[17, 30].

The nonlinear nature of the relation between the glitch properties and the properties of the glitch signature seen in Fig. 4 and Supplementary Fig. 2 makes it difficult to link the two in a simple way when considering the general case of mode coupling in the presence of a core glitch. Nevertheless, in the limit case of no mode coupling, the relation between the glitch position $x^*$ and the period of the glitch signature is straightforward to establish. This limit is a good approximation for the majority of the modes in the oscillation spectra of our targets, since most of the modes are away from the frequencies of maximum coupling. Therefore, we can use this limit to aid our interpretation of the glitch signature seen in Fig. 4 and Supplementary Fig. 2.

In the case of no mode coupling, Eqs. (2) and (4) can be written in terms of the oscillation period of pure gravity modes as

$$P = P_g + \frac{\Delta\Pi_1}{\pi} \arctan\left[ \frac{A \cos(\widetilde{\beta}_1)}{2 + 2A \cos^2(\widetilde{\beta}_2)} \right], \quad (5)$$

where $P_g$ represents the periods of the pure $l = 1$ gravity modes in the absence of the glitch. We can see that the oscillation periods are perturbed by the glitch in a periodic manner, where the period of the signature is given by the period of the cosine function on the nominator of the argument of the arctangent function. Noting that in the absence of mode coupling,

$$\widetilde{\beta}_1 = 2\frac{\widetilde{\omega}_g^*}{2\pi\nu} + 2\varepsilon = 2\pi\frac{\widetilde{\omega}_g^*}{2\pi^2} P + 2\varepsilon, \quad (6)$$

we find that the period of the glitch signature on the mode periods is,

$$\tau_{sig} \equiv \frac{2\pi^2}{\widetilde{\omega}_g^*} = \frac{\Delta\Pi_1}{x^*}. \quad (7)$$

Moreover, given that away from the frequencies where pure acoustic $l = 1$ modes should be situated in the absence of coupling, increments in the stretched periods and in real periods are similar in absolute value $(d\tau \sim −dP)$[18], the period of the glitch signature seen in the stretch period $\tau$ is also given by $\tau_{sig}$. The values of $\tau_{sig}$ are provided in the last column of Table 1 and marked in Fig. 4.

Unfortunately, the arctangent function on the right hand side of eq. (5), prevents us from having a simple relation between the amplitude of the glitch A and the amplitude of the glitch signature. Such simple relation is found when $A \ll 2$, in which case we can approximate the arctangent by its argument and find that the amplitude (maximum value) of the signature is given approximately by

$$A_{sig} = \frac{\Delta\Pi_1}{2\pi} A. \quad (8)$$

In the case of $A \ll 1$, the expression simplifies further, with the signature becoming sinusoidal and given by,

$$P = P_g + \frac{\Delta\Pi_1}{2\pi} A \cos(\widetilde{\beta}_1). \quad (9)$$

In practice, as for most of our targets $A > 2$, the above expression for the amplitude of the signature is not applicable. Indeed, inspection of Supplementary Fig. 2 shows that $A_{sig}$ is generally smaller than predicted by Eq. (8), as expected from Eq. (5) for the values of A found for our targets.

### Stellar models description
In order to verify that the glitches detected in the data are consistent with those expected from the chemical mixing taking place at the edge

of the mixed core, we computed a series of representative stellar models of the clump phase. Using the MESA stellar evolution code[44] we computed nine evolutionary tracks of 1.0, 1.3, and 1.6 $M_\odot$ with metallicity [Fe/H] of −0.25, 0.0, and +0.25 dex. The input physics adopted includes the solar mixtures[45], the radiative opacities from OPAL[46] complemented at low temperatures[47], and the nuclear reaction rates from NACRE[48]. The calculations do not include microscopic diffusion. A step-function overshooting[49] has been used in the clump phase in order to extend the mixed region by a quantity $\alpha_{ovHe} = 0.5 H_p$, where $H_p$ is the pressure scale taken at the classical core-convective border. This scheme guarantees the presence of a discontinuity in the density profile, and therefore in the Brunt-Vaisälä frequency, inside the radiative resonant cavity, necessary for the occurrence of the glitch. We however stopped the computation at the onset of the helium semiconvection, i.e., when the radiative gradient of temperature at the edge of the overall mixed-region rises beyond the adiabatic gradient. In our models, this occurs when about 30–40% of the helium mass fraction in the center has been depleted by the nuclear reactions (depending on the metallicity). This decision has been taken because the correct treatment of the He-semiconvection is still under debate, therefore the internal profiles of the models beyond this point might be affected by uncertainties, with severe consequences to the glitch properties. Along the evolutionary track we saved a series of structure models in the clump phase for which we measured the glitch position and amplitude from the equilibrium structure. The position corresponds to the buoyancy radius at the discontinuity (Fig. 2). The Amplitude is instead calculated using the values of the Brunt-Vaisälä across the discontinuity: $A = (N_{out}/N_{in}) - 1$, where $N$ is the Brunt-Vaisälä frequency, $N_{out}$ is its outer value and $N_{in}$ is its inner value at the glitch position[17].

## Data availability

All data generated in this work are provided within the article as supplementary data files. The raw observational data acquired from archives are available from the corresponding author on reasonable request. Source data are provided with this paper.

## Code availability

The modelling results were provided by the code MESA, available at https://docs.mesastar.org/en/release-r22.05.1/. For the oscillation mode peakbagging, the DIAMONDS code was used (https://diamonds.readthedocs.io/en/latest/). All other codes are available upon request.

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

## Acknowledgements

M.V. acknowledge support from NASA grant 80NSSC18K1582 and is also supported by FEDER - Fundo Europeu de Desenvolvimento Regional through COMPETE2020 - Programa Operacional Competitividade e Internacionalização by these grants: PTDC/FIS-AST/30389/2017 & POCI-01-0145-FEDER-030389. M.S.C. is supported by national funds through FCT in the form of a work contract. P.P.A., M.S.C., and D.B. acknowledge funds from FCT through the research grants UIDB/04434/2020, UIDP/04434/2020 and PTDC/FIS-AST/30389/2017, and from FEDER - Fundo Europeu de Desenvolvimento Regional through COMPETE2020 - Programa Operacional Competitividade e Internacionalização (grant: POCI-01-0145-FEDER-030389).

## Author contributions

M.V. led the project with the help of M.S.C. M.V. worked on extracting the mode parameters from the *Kepler* light-curves with the help of E.C. M.S.C. developed the theoretical description of the discontinuity influence on the mixed-mode pattern with the help of M.V. D.B. performed the stellar modeling. M.V. and P.P.A. worked on the fitting process of the discontinuity. B.M. helped in the identification of stars showing discontinuity signatures.

## Competing interests

The authors declare no competing interests.
