## [Peer Review File · Nature Communications]

REVIEWER COMMENTS

Reviewer #1 (Remarks to the Author):

This paper presents a detailed study of the cores of helium-burning red giants using the technique of asteroseismology. They use data from the Kepler Mission to analyse oscillation frequencies of dipole mixed modes in a sample of 359 red giants. Using methods they have previously published, they find subtle patterns in the frequencies of about 7% of the sample that indicate an abrupt change in structure within the core (called a glitch). By comparing with theoretical models, they conclude that the glitches occur at the interface between the convective and radiative regions in the core. They are not able to identify the reason why the glitches are only seen in a small minority of the sample, and suggest that there may be intermittent changes in the core structures of red giants during their evolution.

Overall, this is a very good paper that presents new and interesting results, and will lead to further work to understand the implications. That work will mainly be theoretical since there is no realistic prospect in the near future for doing better than these four-year Kepler light curves. The methods and results are rather specialised and I am not sure how relevant the paper is for Nature Communications (I would think Nature Astronomy was more appropriate). But either way, this is a fascinating result that deserves to be published rapidly.

It is a puzzle why only 7% of stars show a glitch and I think this requires some revisions in two directions. Firstly, are there really two types of stars, or is there a continuum with a detection threshold? It is hard to say because there is no discussion of the uncertainties in glitch amplitude or of the criteria used to decide between detection and non-detection. Secondly, the properties of the glitch stars could be explored in more detail and compared with the model grid. I list these points in more detail below, as well as a few other suggestions. I found the paper to be extremely well written.

1. There is no discussion of the uncertainties on the glitch parameters, or of the criteria used to decide on which stars have a clear detection. According to ref 16, the values of A and x^* are obtained using MCMC, which can hopefully be used to answer this. They should be able to give uncertainties on amplitude and period for the detected glitches from the posterior distributions. And for the non-glitch stars, there must be an upper limit on glitch amplitude. I would like to see these values listed in the tables and perhaps shown in additional figures.

2. Fig 3 only shows two examples of glitches (are these chosen to be the best ones?!). I suggest showing all 24 glitch stars in supplementary on-line figures. This will be valuable for future workers trying to understand the results.

3. The observed glitches are parametrised by dimensionless parameters A and x^* , which are given in Table 1. Is there a simple way to relate these to the amplitude and period of the glitches shown in Fig. 3, which have units of seconds (or Hz)?

4. The properties of the glitch stars could be explored in more detail. They show them in mass vs $[\text{Fe}/\text{H}]$ in Fig 3, but maybe also show the HR diagram, and the $\Delta \Pi$ vs $\Delta \nu$ diagram. And it would be good to compare somehow with the model results (see next point).

5. Please show more results from the theoretical models. The authors computed 30 different models and measured glitch amplitudes and periods, but all they show from this are the shaded bands in Fig 1. It would be interesting to see these values in other ways. I am not exactly sure how, but I hope the authors can be a bit more inventive. It is a shame to compute all these models and not spend more time comparing with the observational results.

6. Table1: please include values to allow the reader to reproduce Fig 2. That means adding columns with mass and $[\text{Fe}/\text{H}]$. Please also include a similar table in the supplementary material for all the non-glitch stars.

7. The number of self-citations is excessive. Of the 26 references to papers on helio and asteroseismology, half are to papers by the coauthors. Some self-citing is appropriate but the amount seems excessive, especially given the large number of other workers in the field and the existence of several recent and very thorough reviews (none of which are cited).

8. "warrant" should be "guarantee", I think

Reviewer #2 (Remarks to the Author):

Dear Editor(s):

Please find enclosed my referee's report on the paper "Evidence of structural discontinuities in the inner core of red-giant stars", submitted by Mathieu Vrad et al. to Nature Communications.

Using asymptotic formulae, the authors performed a seismological analysis of 359 red giant clump stars using data from the NASA Kepler mission. 24 of these stars showed evidence of core glitches in the g-mode propagation region, and they obtained reliable fits for 23 of these objects. Besides being a validation of their method, this work seeks to set important constraints on the mixing processes taking place in the cores of red giant stars, still an open question in stellar evolution.

I find no obvious weaknesses in the present work, and I appreciate the authors' approach of using asymptotic formulae for the period spacings that have been validated against the results of full numerical models. This approach is more model-independent than just fitting full models to the data, and the procedure is well described in the authors' previous works: Cunha et al. (2015, 2019), and Vrad et al. (2016), Vrad & Cunha (2019). The fact that such glitches are seen in only a fraction of the stars, but these stars are otherwise typical in terms of their parameters, suggests that the glitches are intermittent features that are visible only at particular snapshots in time. Recognition that such stages of evolution exist is important for our detailed understanding of stellar evolution, and asteroseismology offers the only way to constrain these effects. For these reasons, I recommend this paper for rapid publication in Nature Communications.

As a consumer of these and other results, I wonder if there could be a relationship between these glitches and the inferred interior magnetic fields that are believed to lead to a suppression of dipole modes in some stars (e.g., Fuller et al., 2015, Science, 350, 423; Stello et al. 2016, Nature, 529, 364). Of course, these works concentrated on the angular scattering of energy out of $l=1$ modes, but this scattering could have a radial dependence that would lead to ΔP variations as well, although perhaps such structure is thought to be erased during the helium-flash phase. Also, perhaps it is the case that such magnetic features would be much weaker in terms of perturbing N^2 than the μ gradients are, and would likely produce sinusoidal features in ΔP , rather than the very non-sinusoidal features shown here.

I don't know if such a discussion is possible within the limited confines of a Nature Communication, but as (presumably) a member of the target audience, I would find such a short discussion interesting and illuminating.

And, somewhat farther off topic, I wonder if the authors have looked for a suppression of dipole amplitudes in this sample as a possible sign of core magnetic fields, presumably associated with helium burning.

Finally, just a nitpick. The N^2 frequency shown in Fig. 4 is obviously an idealization based on a model with a discontinuous chemical profile. While this is fine to do numerically, physically such an N^2 profile would also contain a delta function, since N^2 depends on the (assumed infinitely sharp) derivative of the chemical profile.

Of course, such a delta function will also produce a feature at the exact same radial location, so modeling both is probably overkill (and inconvenient). However, if it becomes useful to someday make use of the measured glitch height (to try to infer the amount of composition change, say) then including this in the model might be necessary.

Again, I commend the authors for an interesting article and recommend it for rapid publication in Nature Communications.

Reviewer #3 (Remarks to the Author):

Dear Editors and Authors,

I read the paper entitled "Evidence of structural discontinuities in the inner core of red-giant stars" with great interest. If we can indeed obtain a measure of these changes in the structure of the core of the stars that would be very exciting and an important breakthrough.

After reading the paper there are several questions that arose and in my opinion not answered well enough (the authors do touch upon some of the questions listed):

a) how solid are the observational signatures?

b) is the boundary of the convective core the 'only' discontinuity that could cause this?

c) are the parameters of the step function that are used to fit the observations in line with what could be expected from stellar structures?

d) why can it be detected in only a very small sample of stars, while they all have a convective core and thus a convective core boundary?

To start with point a), I tried to reproduce the result shown in Figure 3. In this case I did this for KIC9332840, as there was a KASOC light curve available. I used this light curve and my own tool to perform the computations. A complete reproduction is not possible as not all necessary parameters are provided in the paper. To do so, the coupling parameter q is necessary. Also, I determined the individual frequencies myself as they are not given. A screen shot of my results are attached. In this preliminary analysis, the dipole mode frequencies (red triangles, where the size of the symbol scales with the amplitude of the mode) could also be explained with a slightly different period spacing and rotation. The discrepancy in the results of these two analyses calls for more thorough tests by the authors to firmly show that the observational results they obtain are solid and the variation in the stretched period is indeed due to glitches and cannot be explained by something else, such as rotation as proposed. The authors should also provide the necessary information for the result to be reproduced by others. This is a major issue and needs to be addressed before this work can be recommended for publication.

For point b), I am very surprised that the boundary of the convective core is identified as the 'only' possible glitch in RC stars. When looking at stellar models, I think it is clear that at least in the models remnants of the He-flash could also cause abrupt changes. This should at least be discussed by the authors and shown why they think that the edge of the convective core is indeed the source of the glitch (if it is indeed a glitch we see, as per point a)).

From the text it remains a bit unclear whether or not the fitted parameters for the step at the convective core are in line with model predictions. The authors do address this point, but in my opinion it lacks clarity as it stands.

The last point d) is also addressed and the authors do ask why only so small an amount of stars show a glitch. In my opinion more depth in the discussion is necessary here. Are there theoretical reasons for this, i.e. does the ratio of the wavelength to the spatial extent of the change vary, such that it is no longer a glitch?? Are there observational reasons? If neither of them are present, what else??

Apart from these major points on the content I have a few minor comments:

- bold font part: Is it of importance that the stars in the red clump are the ones with degenerate cores on the RGB? This is not addressed and may be of interest, also given the fact that the He-flash may leave traces...
- “fine structure of stellar interiors”: a bit of an odd phrase as “fine structure” has a particular meaning in atomic physics.
- “seismic mass” may need an explanation for the more general audience.
- “We assess the presence of core glitches after the measurements of the global asteroseismic properties...”: do you mean with “after” that you did this at a later stage, or in the same manner?
- Uncertainties need to be included in the bottom panel of Figure 2.
- The last sentence in the caption of Figure 4 is a bit odd “jump before and after the discontinuity position”?

Please select a star to analyse

KIC 9332840

Choose a page

Stretched Period Echelle

Echelle Parameters

Reset parameters

Save parameters

$\Delta\nu$ (μHz)

4.39

3.95

4.83

$\Delta\Pi_1$ (s)

301.65

-

+

q

0.26

0.00

0.80

ε

-0.05

-1.00

1.00

$\delta\nu_{\text{rot}}$ (μHz)

0.24

0.00

1.20

Stretched Period Echelle Explorer

We thank the referees for the useful and constructive comments on the manuscript. We took them into account. Please find below a detailed description of the different additions to the paper and the detailed answers to your questions.

>Reviewer #1 (Remarks to the Author):

>This paper presents a detailed study of the cores of helium-burning red giants using the technique of asteroseismology. They use data from the Kepler Mission to analyse oscillation frequencies of dipole mixed modes in a sample of 359 red giants. Using methods they have previously published, they find subtle patterns in the frequencies of about 7% of the sample that indicate an abrupt change in structure within the core (called a glitch). By comparing with theoretical models, they conclude that the glitches occur at the interface between the convective and radiative regions in the core. They are not able to identify the reason why the glitches are only seen in a small minority of the sample, and suggest that there may be intermittent changes in the core structures of red giants during their evolution.

>Overall, this is a very good paper that presents new and interesting results, and will lead to further work to understand the implications. That work will mainly be theoretical since there is no realistic prospect in the near future for doing better than these four-year Kepler light curves. The methods and results are rather specialised and I am not sure how relevant the paper is for Nature Communications (I would think Nature Astronomy was more appropriate). But either way, this is a fascinating result that deserves to be published rapidly.

>It is a puzzle why only 7% of stars show a glitch and I think this requires some revisions in two directions. Firstly, are there really two types of stars, or is there a continuum with a detection threshold? It is hard to say because there is no discussion of the uncertainties in glitch amplitude or of the criteria used to decide between detection and non-detection. Secondly, the properties of the glitch stars could be explored in more detail and compared with the model grid. I list these points in more detail below, as well as a few other suggestions. I found the paper to be extremely well written.

Answer: *The answers about the detection limit and the models are presented below. Here we will focus on the question about the continuum between stars with glitch and stars with no glitch.*

The asymptotic expansion can be fitted to the mixed-mode pattern of a lot of objects with very good agreement without glitches. This was proven previously in a large number of studies (Beck et al. 2011, Mosser et al. 2012, Buyschaert et al. 2016). Moreover, considering the errors inferred on the amplitudes of the glitches, we find that in all but two stars with glitch detections, the amplitudes differ from zero by more than 3-sigma, showing that the detection of the glitch signature in those stars is robust. However, there may certainly be some stars that have glitches with low amplitude that we were not able to detect since the error bars on the mode frequencies prevent us to separate those stars from stars with no glitch. From the theoretical point of view, such stars are certainly expected at the beginning of the helium burning phase, where the chemical discontinuity didn't yet have time to build up above the minimum value needed for the glitch to be detected. However, the fact that we find stars where a glitch is not detected spread over the full range of period spacings indicates that a low amplitude glitch at the beginning of helium-core-burning evolution cannot explain the majority of the non-detections. The threshold and the error bars determination are discussed in more detail below.

>1. There is no discussion of the uncertainties on the glitch parameters, or of the criteria used to

decide on which stars have a clear detection. According to ref 16, the values of A and x^* are obtained using MCMC, which can hopefully be used to answer this. They should be able to give uncertainties on amplitude and period for the detected glitches from the posterior distributions. And for the non-glitch stars, there must be an upper limit on glitch amplitude. I would like to see these values listed in the tables and perhaps shown in additional figures.

Answer: Concerning the uncertainties, they are obtained from the MCMC fitting and they are given in Table 1 of the paper for the amplitude (A) and period (x^*) of the glitch modulation.

Concerning the criteria used to decide which stars have a clear glitch detection or not, this was determined through the frequency deviations from the expected mixed mode pattern for each star. If a star shows a high number of mode frequencies with an important deviation from the asymptotic mixed-mode pattern (taking into account the uncertainties on the frequencies), then the star is considered to be a candidate for possessing gravity glitches. The stars are then checked one by one in order to see if a better solution with different parameters ($\delta\pi, q$) or the presence of rotation can explain those deviations. If no satisfactory explanation is present, the star is considered to have a glitch. A description of this selection was added to the additional materials in the paper.

2. Fig 3 only shows two examples of glitches (are these chosen to be the best ones?!). I suggest showing all 24 glitch stars in supplementary on-line figures. This will be valuable for future workers trying to understand the results.

Answer: We added the additional figures.

3. The observed glitches are parametrised by dimensionless parameters A and x^* , which are given in Table 1. Is there a simple way to relate these to the amplitude and period of the glitches shown in Fig. 3, which have units of seconds (or Hz)?

Answer: The relation between the parameters of the glitch that characterize the structural change and the amplitude and period of the glitch signature is not straightforward in the general case, where coupling and glitch effects are simultaneously present. However, there is a limit case that can be considered and is still quite informative. That is the limit of no coupling. Because the coupling is important only near the pure acoustic-mode frequencies, neglecting it still provides a good indication of how the signature behaves away from those frequencies and, in particular, a measure of the period of the glitch signature.

Moreover, while in most cases our glitches are not small (as seen by the non-sinusoidal behavior of the signature), if, in addition, we consider the small glitch limit, we can understand how the amplitudes of the glitch and that of the glitch signature are related in that limit case and how that relation is modified when the glitch amplitude is not small. In particular, the amplitude of the signature derived in this limit is found to be larger than the real one, as expected from comparison of the limit case expression with the general expression.

We have added a brief discussion of the no coupling case and small glitch limit to the manuscript. We have also added the period of the signature derived in the no coupling case to the table, so that it can be compared to that seen in Fig 3. We also changed Fig 3, so that the

vertical axis is the period, rather than frequency, so that the signature is periodic and the relation with the period of the signature now derived can be readily seen in the figure

4. The properties of the glitch stars could be explored in more detail. They show them in mass vs [Fe/H] in Fig 3, but maybe also show the HR diagram, and the Delta Pi vs Delta nu diagram. And it would be good to compare somehow with the model results (see next point).

Answer: The Figure 3 was modified in order to add the DeltaPi vs Deltanu diagram. However, all stars we considered are low-mass clump objects that have similar effective temperature and Log(g), thus their position in the HR diagram doesn't bring a lot of information. Therefore, we decided to not add the HR diagram to Fig. 3.

5. Please show more results from the theoretical models. The authors computed 30 different models and measured glitch amplitudes and periods, but all they show from this are the shaded bands in Fig 1. It would be interesting to see these values in other ways. I am not exactly sure how, but I hope the authors can be a bit more inventive. It is a shame to compute all these models and not spend more time comparing with the observational results.

Answer: In order to have a more comprehensive view of the model results, we have computed a number of additional evolution sequences by changing the stellar mass and metallicity. While a direct comparison with the data is still hampered by the fact that the model sequences only cover the first half of the he-core evolution, we added a couple of panels to Fig. 1 showing the glitch properties inferred from these sequences of models and discussed the matter in more detail. In particular, the new results show that the glitch related to the chemical discontinuity evolves along the clump phase towards higher glitch amplitude and lower glitch position in the radiative resonant cavity. The glitch properties also highly depend on stellar metallicity which can also contribute to a significant dispersion in the observed glitch properties.

6. Table1: please include values to allow the reader to reproduce Fig 2. That means adding columns with mass and [Fe/H]. Please also include a similar table in the supplementary material for all the non-glitch stars.

Answer: We added a new table with the requested information for all stars in the sample.

7. The number of self-citations is excessive. Of the 26 references to papers on helio and asteroseismology, half are to papers by the coauthors. Some self-citing is appropriate but the amount seems excessive, especially given the large number of other workers in the field and the existence of several recent and very thorough reviews (none of which are cited).

Answer: Following the comments, some citations were replaced, and others were added, including a recent review.

8. "warrant" should be "guarantee", I think

Answer: *The word has been corrected.*

Reviewer #2 (Remarks to the Author):

Dear Editor(s):

Please find enclosed my referee's report on the paper "Evidence of structural discontinuities in the inner core of red-giant stars", submitted by Mathieu Vrad et al. to Nature Communications.

Using asymptotic formulae, the authors performed a seismological analysis of 359 red giant clump stars using data from the NASA Kepler mission. 24 of these stars showed evidence of core glitches in the g-mode propagation region, and they obtained reliable fits for 23 of these objects. Besides being a validation of their method, this work seeks to set important constraints on the mixing processes taking place in the cores of red giant stars, still an open question in stellar evolution.

I find no obvious weaknesses in the present work, and I appreciate the authors' approach of using asymptotic formulae for the period spacings that have been validated against the results of full numerical models. This approach is more model-independent than just fitting full models to the data, and the procedure is well described in the authors' previous works: Cunha et al. (2015, 2019), and Vrad et al. (2016), Vrad & Cunha (2019). The fact that such glitches are seen in only a fraction of the stars, but these stars are otherwise typical in terms of their parameters, suggests that the glitches are intermittent features that are visible only at particular snapshots in time. Recognition that such stages of evolution exist is important for our detailed understanding of stellar evolution, and asteroseismology offers the only way to constrain these effects. For these reasons, I recommend this paper for rapid publication in Nature Communications.

As a consumer of these and other results, I wonder if there could be a relationship between these glitches and the inferred interior magnetic fields that are believed to lead to a suppression of dipole modes in some stars (e.g., Fuller et al., 2015, Science, 350, 423; Stello et al. 2016, Nature, 529, 364). Of course, these works concentrated on the angular scattering of energy out of $l=1$ modes, but this scattering could have a radial dependence that would lead to ΔP variations as well, although perhaps such structure is thought to be erased during the helium-flash phase. Also, perhaps it is the case that such magnetic features would be much weaker in terms of perturbing N^2 than the μ gradients are, and would likely produce sinusoidal features in ΔP , rather than the very non-sinusoidal features shown here.

I don't know if such a discussion is possible within the limited confines of a Nature Communication, but as (presumably) a member of the target audience, I would find such a short discussion interesting and illuminating.

Answer: *We thank the referee for the discussion on this subject. The presence of interior magnetic fields as inferred by Fuller et al. (2015) and Stello et al. (2016) leads to the damping of the $l=1$ oscillations which is not seen in the objects we considered. They, on the contrary, have very clear dipole oscillations. The recent work performed by Bugnet et al. (2021) allows also to see that the deviations from the classical mixed-mode pattern due to high magnetic field is very different from what is predicted for core glitches, like in Cunha et al. (2019), and what was observed in this work.*

And, somewhat farther off topic, I wonder if the authors have looked for a suppression of dipole

amplitudes in this sample as a possible sign of core magnetic fields, presumably associated with helium burning.

Answer: As stated above, the amplitude of the dipole modes of this sample are not damped which allowed us to clearly see the gravity-dominated mixed modes of the different stars, therefore permitting us to obtain a measurement on the glitch parameters.

Finally, just a nitpick. The N^2 frequency shown in Fig. 4 is obviously an idealization based on a model with a discontinuous chemical profile. While this is fine to do numerically, physically such an N^2 profile would also contain a delta function, since N^2 depends on the (assumed infinitely sharp) derivative of the chemical profile.

Of course, such a delta function will also produce a feature at the exact same radial location, so modeling both is probably overkill (and inconvenient). However, if it becomes useful to someday make use of the measured glitch height (to try to infer the amount of composition change, say) then including this in the model might be necessary.

Answer: We fully agree. However, comparison of the results for a glitch modelled by a Dirac delta function and a glitch modelled by a step function, studied by Cunha et al. 2015 and Cunha et al., 2019, respectively, shows that the main impact of the form of the glitch is to change the frequency dependence of the signature's amplitude. Given the limited extent of mode frequencies available for each star, it is difficult to determine the frequency dependence of the amplitude, and establish with confidence the form of the glitch on N^2 . But we agree that the interpretation of the glitch amplitude depends on the form of the glitch and that further comparison with different glitch models should be considered in future work.

Again, I commend the authors for an interesting article and recommend it for rapid publication in Nature Communications.

Reviewer #3 (Remarks to the Author):

Dear Editors and Authors,

I read the paper entitled "Evidence of structural discontinuities in the inner core of red-giant stars" with great interest. If we can indeed obtain a measure of these changes in the structure of the core of the stars that would be very exciting and an import breakthrough.

After reading the paper there are several question that arose and in my opinion not answered well enough (the authors do touch upon some of the questions listed):

- a) how solid are the observational signatures?
- b) is the boundary of the convective core the 'only' discontinuity that could cause this?
- c) are the parameters of the step function that are used to fit the observations in line what could be expected from stellar structures?
- d) why can it be detected in only a very small sample of stars, while they all have a convective core and thus a convective core boundary?

To start with point a), I tried to reproduce the result shown in Figure 3. In this case I did this for KIC9332840, as there was a KASOC light curve available. I used this light curve and my own tool to perform the computations. A complete reproduction is not possible as not all necessary parameters

are provided in the paper. To do so, the coupling parameter q is necessary. Also, I determined the individual frequencies myself as they are not given. A screen shot of my results are attached. In this preliminary analysis, the dipole mode frequencies (red triangles, where the size of the symbol scales with the amplitude of the mode) could also be explained with a slightly different period spacing and rotation. The discrepancy in the results of these two analyses calls for more thorough tests by the authors to firmly show that the observational results they obtain are solid and the variation in the stretched period is indeed due to glitches and cannot be explained by something else, such as rotation as proposed. The authors should also provide the necessary information for the result to be reproduced by others. This is a major issue and needs to be addressed before this work can be recommended for publication.

Answer: We thank the referee for his thorough verification of our data and results. To ease the reproduction of the results in an easier way, we furnished the coupling parameter in Table 1 and the mixed-mode frequencies attached to this answer. These frequencies will be released with the article. We tried to reproduce the referee's results with the proposed parameters (see attached Figure pb9332840_rot) but it doesn't appear to give a satisfactory fit in the echelle diagram for the full spectra. As a side note, it's important to also note that the rotation value proposed by the referee is more than very high for a clump object. If we follow the literature, it was never observed before (e.g. Mosser et al. 2012), it would therefore be an important discovery. It's very unlikely that this situation is present for each star we present in this work.

For point b), I am very surprised that the boundary of the convective core is identified as the 'only' possible glitch in RC stars. When looking at stellar models, I think it is clear that at least in the models remnants of the He-flash could also cause abrupt changes. This should at least be discussed by the authors and shown why they think that the edge of the convective core is indeed the source of the glitch (if it is indeed a glitch we see, as per point a)).

Answer: The boundary of the convective core is not the only possible glitch in RC stars. It has been shown that the H-burning shell can also produce a glitch in specific circumstances in low mass stars, in particular between the He-flash and the start of quiet helium-core burning (Cunha et al., 2015, Deheuvels&Belkacem 2018). However, that is found not to be the case for low mass stars during the helium-core burning phase (Bossini et al, 2015; Cunha et al., 2015). In addition, even if the H-shell were a glitch during the clump phase (i.e., if it were narrower than the local wavelength) it's position in the propagation cavity would cause a signature with a much larger associated x^ , than what is observed in this work (Deheuvels&Belkacem 2018). What is argued here is that the only glitch that can correspond to the observations is the one associated with the boundary of the convective zone. The possibility of a glitch associated to the He-flash remnants was also investigated; however, according to models the flashes seem to leave few chemical gradient traces on low mass objects even during the flashes (see e.g., Deheuvels&Belkacem 2018). The more evolved clump models seem also to not show any significant discontinuity due to the previous helium flashes (Bossini et al. 2017). An example of the discontinuities left from the helium flash in one of our models has been added in the paper. A discussion on this was added in the main article.*

From the text it remains a bit unclear whether or not the fitted parameters for the step at the convective core are inline with model predictions. The authors do address this point, but in my opinion it lacks clarity as it stands.

Answer: The discussion on the model predictions was extended in the main text and new panels illustrating the model glitch properties were added to Fig. 1. In particular, we have computed a number of new evolution sequences, by varying the stellar mass and metallicity, so that a better comparison with the observational results becomes possible.

The last point d) is also addressed and the authors do ask why only so small an amount of stars show a glitch. In my opinion more depth in the discussion is necessary here. Are there theoretical reasons for this, i.e. does the ratio of the wavelength to the spatial extent of the change vary, such that it is no longer a glitch?? Are there observational reasons? If neither of them are present, what else??

Answer: In our opinion, the fact that no systematic differences are found between the sample of stars where glitches are detected and the whole sample is a key result of our study which may influence future theoretical developments. This dichotomy could result from the additional action of physical process (e.g., an instability or a smoothing process) capable of either temporarily removing the structural variation or smoothing it (so that it is no longer a glitch). However, as explained in the section "methods", there is still a significant degree of uncertainty in the models of clump stars, especially in what concerns the treatment of He semiconvection. As such, we opted not to dwell about the possible theoretical reasons for the observed dichotomy (which could be seen as speculation), but simply argue that the observed dichotomy is a robust result.

We have now extended a bit that discussion, in particular by stating explicitly that the cause cannot be associated to differences in the quality of the data of stars with and without glitch detection, nor to a systematic difference in their evolutionary status.

Apart from these major points on the content I have a few minor comments:

- bold font part: Is it of importance that the stars in the red clump are the ones with degenerate cores on the RGB? This is not addressed and may be of interest, also given the fact that the He-flash may leave traces...

Answer: The selection of clump low-mass stars enables us to obtain a sample of stars that follow similar evolutionary tracks during the clump phase, having ignited helium in their core under electron degeneracy, therefore having the same helium-core. This selection is, thus, expected to suppress somewhat the spread in the glitch properties during the clump phase by having stars that start the helium burning phase with the same core mass. In fact, inspection of the new panels in Fig. 1 shows that indeed the model-glitch properties are not expected to depend significantly on stellar mass, in this mass range. This choice doesn't mean that there are no core glitches in the secondary clump or other red giant objects.

- "fine structure of stellar interiors": a bit of an odd phrase as "fine structure" has a particular meaning in atomic physics.

Answer: It has been changed.

- “seismic mass” may need an explanation for the more general audience.

Answer: It has been modified.

- “We assess the presence of core glitches after the measurements of the global asteroseismic properties...”: do you mean with “after” that you did this at a later stage, or in the same manner?

Answer: What we meant is that the measurement of the global seismic parameters was performed first, before the glitch characterization. The sentence has been modified.

- Uncertainties need to be included in the bottom panel of Figure 2.

Answer: An indication of the typical value of the uncertainties was added in Figure 2.

- The last sentence in the caption of Figure 4 is a bit odd “jump before and after the discontinuity position”?

Answer: It has been modified.

9332840 - Dnu = 4.39 muHz - T = 301.50 s - q = 0.26 - nurot = 240 nHz

REVIEWER COMMENTS

Reviewer #1 (Remarks to the Author):

I thank the authors for responding so carefully to the comments. I recommend that the revised paper is accepted. My only small suggestion concerns the new lower panels of Fig 1, which rely on the reader having good colour vision. It would help to use different symbol shapes to distinguish the various groups of points.

Reviewer #2 (Remarks to the Author):

I agree with the responses the authors have made to the reviewer comments and the corresponding changes they have made to the manuscript. I recommend this article for publication.

Reviewer #3 (Remarks to the Author):

The modified version of the paper “Evidence of structural discontinuities in the inner core of red-giant stars” is certainly improved compared to the earlier version. However, in my view for this new paradigm to be presented in a convincing way, it is not worked through well enough. To really make a mark this is necessary in my opinion. Here I provide suggestions to do so both on the theoretical as well as on the observational front. Note though that this may not be complete.

It is described how the frequencies are detected and fitted, however there is no figure that shows this. For the current work, I think it would be essential to also show a power density spectrum of one of the stars and indicate which peaks are chosen. In this way figure 3 can be put much more in context and give more meaning to the data points. Also the radial modes could be indicated in Figure 3 to give an idea of how many radial orders the period spreads over.

As a second point, I think it should be clarified how the acoustic radius and amplitude of the glitch are determined for the models. In the current version it is unclear to me if indeed oscillations for the models were computed and the glitch was measured from there, or if they were derived from the

Brunt-Vaisälä frequency directly. I think the first method should be used and may also provide some indications on why the glitch is visible in 7% of the stars and not in the other stars. When computing the frequencies, it would also be good to compare stretched period echelle diagrams for the oscillations of a model with a radiative and a model with an adiabatic temperature gradient in the core overshoot.

The comment in your response on the rotation of 0.24uHz may well be true as the rotation rate of CHeB stars is generally low. What would a typical value be? In Figure 3, no ridges due to rotation are visible. Is rotation not measured in these stars? Are we looking at the $m=0$ ridge only? How about the other stars in the sample? Do none of them have rotation? Could the authors elaborate on how rotation was handled.

To put the 7% number in context it should be explained how the 359 stars were selected. Is this a proper representation of clump stars? Kepler observed more stars in the clump, so why were these selected? This should be clarified.

On the comment regarding the signatures from the He-flash. Did the authors make sure that no smoothing was applied to the Brunt-Vaisälä frequency?? In many stellar evolution codes smoothing is a standard procedure and I think it should be checked and stated that this is not the case here. (This also comes back to my earlier point on whether frequencies are calculated for the models, as smoothing would have an impact there too).

The authors now checked the impact of $\Delta\pi$, q and rotation on the stretched period echelle diagram. However, what are the stepsize and ranges for which this was checked? Furthermore, $d01$ is not mentioned. It is known that this will also have an effect. How did the authors deal with that?

In some areas of the paper some terms are used that I didn't find clear immediately "the frequency of maximum coupling" is 'closest to the nominal acoustic dipole mode' meant here?

The information in Table 2 is very much appreciated. Good scientific practise requires that uncertainties on values should also be quoted. Same for a few columns in Table 1.

Towards the bottom of page 5: The reference should be to Figure 1 and not to Figure 2.

Personally, I would not dwell in the paper on the mean molecular weight discontinuity at the base of the convection zone, as this is a feature on the RGB and not present in clump stars in the mass range selected. But that is a matter of preference and does not influence any recommendation on the publication.

We thank the referees for their constructive comments. We took them into account as best as we could and modified the paper accordingly. Nevertheless, in our opinion, one of the main suggestions cannot be properly addressed with the current knowledge in the field of clump star models. Please find below the answer to the comments with further discussion on the issue mentioned above with supporting references.

Reviewer #1 (Remarks to the Author):

I thank the authors for responding so carefully to the comments. I recommend that the revised paper is accepted. My only small suggestion concerns the new lower panels of Fig 1, which rely on the reader having good colour vision. It would help to use different symbol shapes to distinguish the various groups of points.

***Answer:** Figure was modified accordingly.*

Reviewer #2 (Remarks to the Author):

I agree with the responses the authors have made to the reviewer comments and the corresponding changes they have made to the manuscript. I recommend this article for publication.

***A:** Many thanks for the constructive comments and the positive report.*

Reviewer #3 (Remarks to the Author):

The modified version of the paper "Evidence of structural discontinuities in the inner core of red-giant stars" is certainly improved compared to the earlier version. However, in my view for this new paradigm to be presented in a convincing way, it is not worked through well enough. To really make a mark this is necessary in my opinion. Here I provide suggestions to do so both on the theoretical as well as on the observational front. Note though that this may not be complete.

***A:** Many thanks for the constructive comments and suggestions which we address below.*

It is described how the frequencies are detected and fitted, however there is no figure that shows this. For the current work, I think it would be essential to also show a power density spectrum of one of the stars and indicate which peaks are chosen. In this way figure 3 can

be put much more in context and give more meaning to the data points. Also the radial modes could be indicated in Figure 3 to give an idea of how many radial orders the period spreads over.

A: An additional Figure was added showing the oscillation spectrum of star KIC3544063 with the fitting result frequencies of the peaks identified as mixed-modes highlighted. Figure 3 was also updated following the suggestion: the periods of the radial orders were put forward.

As a second point, I think it should be clarified how the acoustic radius and amplitude of the glitch are determined for the models. In the current version it is unclear to me if indeed oscillations for the models were computed and the glitch was measured from there, or if they were derived from the Brunt-Vaisälä frequency directly. I think the first method should be used and may also provide some indications on why the glitch is visible in 7% of the stars and not in the other stars. When computing the frequencies, it would also be good to compare stretched period echelle diagrams for the oscillations of a model with a radiative and a model with an adiabatic temperature gradient in the core overshoot.

A: The buoyancy radius and amplitude of the glitch were determined directly from the equilibrium structure. We added a statement to the new version to clarify that this was the method adopted. We understand the referee's concerns about the method we used, however we kept that option in the new version for the reasons enumerated below.

1) This work presents new observational results which we hope will help improve state-of-the art models. However, there is no aim (nor is it currently possible, as explained in point 2) to perform a detailed data-model comparison. Hence, models are used here only to qualitatively support the interpretation of the observational results. For that purpose, inferring the glitch properties from the equilibrium models is sufficient. Although not aimed specifically at clump stars, several works published in the literature show that the properties of acoustic and buoyancy glitches inferred directly from the equilibrium structure are in reasonable agreement with those inferred from fitting the mode frequencies, minus some small bias, which is irrelevant for the qualitative support required here, [e.g., see figure attached from Cunha et al. 2019, MNRAS 490, 909 for a buoyancy glitch in a 6 Msun main-sequence star with a similar functional form as those in our work (step function); note the similarity of the full black line in the lower panel (glitch seen directly in the buoyancy frequency of the model) and the grey dashed line (glitch recovered using the parameters inferred from fitting the seismic data produced for the model)].

2) We agree that if the aim were to perform a detailed data-model comparison (e.g., to determine the best model) the models should be treated exactly as the observed data, which would imply computing the model frequencies and inferring the glitch properties from those frequencies. Unfortunately, models of low-mass clump objects are not sufficiently good to perform such an exercise. Firstly, as explained in the paper (section methods), there is significant debate concerning the correct treatment of the He-semiconvection, which led us to stop the evolution of our models midway through the core-helium burning phase. Therefore,

our models cannot fully cover the same parameter space as the observed stars. Secondly, while we may rely on the equilibrium models up to the onset of He-semiconvection to estimate the position and amplitude of the glitch, the seismic properties computed from these equilibrium models (in particular the individual mixed-mode frequencies) are hardly trustworthy. In fact, there is an ongoing community effort to improve models of red giants aiming at a model-data comparison based on individual non-radial modes, but there is still some way to go (see, e.g., the abstract of the paper Christensen-Dalsgaard, et al., 2020; [10.1051/0004-6361/201936766](https://arxiv.org/abs/10.1051/0004-6361/201936766)). Given the current state-of-the-art, we thus consider that it is more reliable to infer the properties of the glitches from equilibrium structure and we emphasize that such inference is sufficient for the goals of the paper.

The comment in your response on the rotation of 0.24 μ Hz may well be true as the rotation rate of CHeB stars is generally low. What would a typical value be? In Figure 3, no ridges due to rotation are visible. Is rotation not measured in these stars? Are we looking at the $m=0$ ridge only? How about the other stars in the sample? Do none of them have rotation? Could the authors elaborate on how rotation was handled.

A: *The typical value of rotation for this type of stars (low-mass clump objects) is between 0.02 μ Hz and 0.10 μ Hz, following for example Figure 6 of Mosser et al. (2012); [1209.3336.pdf \(arxiv.org\)](https://arxiv.org/abs/1209.3336). This type of low rotation does not allow to have crossed mixed-modes frequencies of different degree l . This type of behavior was never observed for clump objects.*

The rotation was indeed taken into account at the moment the mode identification was realized. A smoothing of the spectra with a low-pass filter with a width equal to $\Delta\nu/100$ was performed. Then, for each mode, we took the threshold ratio used to determine the peak significance and verified if the smoothed spectra showed several peaks instead of one above that threshold. If that was not the case, only one mode was fitted to the peak. Otherwise, the number of fitted mode corresponded to the number of peaks above the threshold. Most of the stars for which a glitch was detected didn't show any rotation following that method. Supplementary informations to the Methods section was added to describe the way mode splitting was taken into account.

To put the 7% number in context it should be explained how the 359 stars were selected. Is this a proper representation of clump stars? Kepler observed more stars in the clump, so why were these selected? This should be clarified.

A: *The sample was picked from the sample for which a determination of the gravity mode period spacing was previously performed in Vrad et al. 2016 ([1602.04940.pdf \(arxiv.org\)](https://arxiv.org/abs/1602.04940)). The stars were selected following two major criteria: we selected low-mass objects ($M < 1.9M_{\odot}$) in order to avoid the stars that start the He-burning phase with different He-core masses. We selected also stars for which a period-spacing was previously measured with different codes in the work of Vrad et al. (2016) to have high signal-to-noise spectra where mixed-modes are well visible. The final sample was however still too large to check the results individually, so we took the first stars in the sample by KIC number to perform a thorough analysis. A statement was added in the methods part concerning that last point.*

On the comment regarding the signatures from the He-flash. Did the authors make sure that no smoothing was applied to the Brunt-Vaisälä frequency?? In many stellar evolution codes smoothing is a standard procedure and I think it should be checked and stated that this is not the case here. (This also comes back to my earlier point on whether frequencies are calculated for the models, as smoothing would have an impact there too).

A: The models were not smoothed. The source of the confusion may come from the fact that double points were added in the locations of the He-flash signatures. Besides that, smoothing was not performed on the Brunt-Vaisälä frequency.

The authors now checked the impact of $\Delta\pi$, q and rotation on the stretched period echelle diagram. However, what are the stepsize and ranges for which this was checked? Furthermore, d_{01} is not mentioned. It is known that this will also have an effect. How did the authors deal with that?

A: When the $\Delta\pi$ and q parameters are measured, it's possible that less probable solutions are also highlighted, for example some aliases (see for example Figure 5 and the discussion of part 3.5.2 of: 1602.04940.pdf (arxiv.org)) that result from the absence of some gravity-dominated mixed modes that does not appear because of their high inertia. Those aliases are typically present in the range of 10% around the measured $\Delta\pi$ value which was thus the range of values that were selected with a step of 0.1s. The coupling parameter value range included the typical range of that parameter for a clump star: between 0.05 and 0.5 with step of 0.05.

The d_{01} is indeed an important parameter. This parameter was derived from the universal pattern as presented in table 2 of Mosser et al. (2011: 1011.1928.pdf (arxiv.org)). We checked before for a few objects that this parameter does not affect the glitch deviations. This parameter has in fact no influence on gravity-dominated mixed-modes compared to pressure-dominated mixed-modes (e.g. Figure 7 of: 1509.06193.pdf (arxiv.org)). Therefore, the influence of the d_{01} on the global mixed-mode pattern is weak and will not affect long glitch perturbations like the ones we are putting forward. In order to be sure of that, we performed another thorough check by modifying the d_{01} parameter by a value of +/-5% of the large separation (D_{nu}) with a stepsize of 1%. This check allowed us to say that no alternative solutions can be found to a glitch by modifying the d_{01} parameter. The text of the paper was modified accordingly.

In some areas of the paper some terms are used that I didn't find clear immediately "the frequency of maximum coupling" is 'closest to the nominal acoustic dipole mode' meant here?

A: This sentence was indeed not clear. Here, we wanted to highlight the considered frequencies that are away from the pure acoustic $l=1$ modes that would be observed if no coupling were present. The sentence was modified in order to clarify the statement.

The information in Table 2 is very much appreciated. Good scientific practise requires that uncertainties on values should also be quoted. Same for a few columns in Table 1.

A: The uncertainties were added on the different columns.

Towards the bottom of page 5: The reference should be to Figure 1 and not to Figure 2.

A: It has been corrected.

Personally, I would not dwell in the paper on the mean molecular weight discontinuity at the base of the convection zone, as this is a feature on the RGB and not present in clump stars in the mass range selected. But that is a matter of preference and does not influence any recommendation on the publication.

A: We thank the referee for the pertinent comment, but we think it is worth mentioning since it can be present for some secondary clump stars with higher mass than our sample. Therefore, we decided to keep that reference in the manuscript

REVIEWER COMMENTS

Reviewer #3 (Remarks to the Author):

I thank the authors for including my suggestions and clarifying some of my questions. The addition of the uncertainties and the additional figure also raise some questions:

- how are the uncertainties computed? An uncertainty of a couple of seconds on the period spacing is realistic. What about the stars where the uncertainty is larger than 10-20 seconds?? It is known that all other values are correlated with this and that the result is very sensitive to the exact value of the period spacing, i.e. a change of the order of 0.1 second in period spacing can already have a significant impact on the stretched period echelle, see the attached figure, courtesy to one of my MSc students working on rotation. The black dots are the "baseline" and the other colours show the stretched period echelle with the period spacing changed by 0.2 seconds. I think information on the computation of the uncertainties is to show that it is indeed possible to get the glitch parameters as accurate as claimed given the uncertainties on the frequencies and other parameters.

My second point is on the added figure. For this star, both in the lowest as well as in the highest acoustic order there are mixed modes that are / should be there, and not picked up. What is the impact of these missing modes on the result?

Artificial peaks, $\Delta\Pi_1^* = 75\text{s}$ 
We thank the third referee for the comments on the modifications that were done to the paper. Please find below our answers to those comments with a few additional Figures.

Reviewer #3 (Remarks to the Author):

I thank the authors for including my suggestions and clarifying some of my questions. The addition of the uncertainties and the additional figure also raise some questions:

- how are the uncertainties computed? An uncertainty of a couple of seconds on the period spacing is realistic. What about the stars where the uncertainty is larger than 10-20 seconds?? It is known that all other values are correlated with this and that the result is very sensitive to the exact value of the period spacing, i.e. a change of the order of 0.1 second in period spacing can already have a significant impact on the stretched period echelle, see the attached figure, courtesy to one of my MSc students working on rotation. The black dots are the 'baseline' and the other colours show the stretched period echelle with the period spacing changed by 0.2 seconds. I think information on the computation of the uncertainties is to show that it is indeed possible to get the glitch parameters as accurate as claimed given the uncertainties on the frequencies and other parameters.

The uncertainties are computed following Appendix A of Vrad et al. (2016: 1602.04940.pdf (arxiv.org)). The larger uncertainties found with this method translate the possible confusion with an alias in the Fourier spectrum of the modified oscillation spectrum frequencies. In those cases, we checked independently each star and the presence of aliases was ruled out in all stars exhibiting glitches, as well as in most other stars in the sample. We added a statement in the paper to clarify the goal of that verification. Since these verifications were performed, it is not necessary to put forward the uncertainties related to aliases when they have been ruled out. We therefore adjusted the uncertainties in Tables 1 and 2 to reflect that check, keeping the large uncertainties only in the case of a few stars in Table 2 for which aliases cannot be ruled out.

It is also important to mention that the large uncertainties present in Table 1 for the period spacings translate the difficulty of the measurement for stars showing important glitches. Glitches correspond indeed to a modulation of the period spacing, therefore adding a supplementary difficulty for the measurement.

My second point is on the added figure. For this star, both in the lowest as well as in the highest acoustic order there are mixed modes that are / should be there, and not picked up. What is the impact of these missing modes on the result?

The modes that were not selected correspond to the ones that do not satisfy the criteria described in the data preparation description. These peaks did not reach the threshold value, for the smooth oscillation spectrum, to guarantee the rejection of the pure noise hypothesis. To see the impact of lowering the threshold, we performed the fitting including these modes for the star KIC3544063. The comparative results are shown in the two Figures below. The top one shows the original fitting as described in the paper, the bottom one corresponds to the fit with the addition of the supplementary modes. The period and phase of the glitch modulation agree within the errors ($x^ = 0.036 \pm 0.001$ and $\varepsilon = 0.277 \pm 0.043$ for the original fit, $x^* = 0.034 \pm 0.003$ and $\varepsilon = 0.214 \pm 0.023$ for the new fit), whereas its amplitude was more significantly affected ($A = 5.097 \pm 0.542$ for the original fit, $A = 7.697 \pm 1.199$). This is likely caused by the presence of the additional frequencies that are allowing a better coverage of the modulation thus pushing the modulation amplitude parameter towards higher values. In any case, even if some specific values may change slightly when including these peaks, the overall analysis and conclusions would not be modified.*

REVIEWERS' COMMENTS

Reviewer #3 (Remarks to the Author):

Dear authors,

many thanks for taking care of my concerns. I think the results (including the uncertainties) now look more convincing. I have no further comments.

Kind regards.